# Ring Attractors as the Basis of a Biomimetic Navigation System

**DOI:** 10.3390/biomimetics8050399

**Published:** 2023-09-01

**Authors:** Thomas C. Knowles, Anna G. Summerton, James G. H. Whiting, Martin J. Pearson

**Affiliations:** 1Bristol Robotics Laboratory, University of the West of England, Bristol BS16 1QY, UK; martin.pearson@brl.ac.uk; 2School of Engineering, University of the West of England, Bristol BS16 1QY, UK; anna.summerton@uwe.ac.uk (A.G.S.); james.whiting@uwe.ac.uk (J.G.H.W.)

**Keywords:** ring attractors, navigation, grid cells, predictive coding, NEST

## Abstract

The ability to navigate effectively in a rich and complex world is crucial for the survival of all animals. Specialist neural structures have evolved that are implicated in facilitating this ability, one such structure being the ring attractor network. In this study, we model a trio of Spiking Neural Network (SNN) ring attractors as part of a bio-inspired navigation system to maintain an internal estimate of planar translation of an artificial agent. This estimate is dynamically calibrated using a memory recall system of landmark-free allotheic multisensory experiences. We demonstrate that the SNN-based ring attractor system can accurately model motion through 2D space by integrating ideothetic velocity information and use recalled allothetic experiences as a positive corrective mechanism. This SNN based navigation system has potential for use in mobile robotics applications where power supply is limited and external sensory information is intermittent or unreliable.

## 1. Introduction

To sense and act upon the world in an adaptive way, the brain must model the relevant dynamics of the world, enabling itself to anticipate and proactively engage with events, rather than being restricted to a loop of stimulus-response. This is aided by implicit neural structures with dynamics that reflect some of the physical properties inferred from the external world. One system in particular that is common across many species and genera is the ring attractor. Found as physical rings of neurons in flies and locusts [1] and virtual rings in mammals [2,3], this neural structure allows for the direct modelling of a cyclic physical variable—for example head direction—that can accept contributions from multiple information sources whilst obeying physical constraints.

Physically-constrained problem spaces ensure the responses of an organism account for its physical capabilities, particularly its limited repertoire of sensory and motor systems, and its inability to act instantaneously. Examples of problem spaces that are intrinsically physical include episodic memory (reconstructing a linear event narrative from memory snapshots), communication (be it with speech, gesture or dance) and moving purposefully from place to place within the world; the domain of navigation.

Extensively studied in both biology and robotics, it has long been established that two of the key tasks of navigation—building a representation of the environment, and locating oneself within that environment—must be solved simultaneously, an approach popularly referred to in robotics as Simultaneous Localisation and Mapping (SLAM) [4]. Typical state-of-the-art SLAM algorithms use a variety of proven techniques, be it finding key-points in a visual scene [5], building graphs relating visited locations [6], and include techniques like dead-reckoning to move through areas with limited sensory richness [7]. Robotic SLAM algorithms make use a variety of sensors, including cameras, LIDAR and GPS signals. Bio-inspired and bio-plausible approaches have also been proposed and evaluated [8,9,10], often with the intention of leveraging the robustness and power efficiency of spiking hardware.

Whilst mobile robots need to be able to navigate effectively, they must also work within the constraints of their designs, particularly in regards to their power source. SLAM systems can be notoriously power-hungry, particularly when LIDAR sensors are required [11], whereas the human brain, a capable and robust navigation system, requires a mere 20 Watts of power [12]. This motivates a biomimetic approach to designing navigation systems for power-efficient mobile robots, inspired by neuroscience and able to take advantage of recently available neuromorphic hardware [13].

Within mammalian neuroscience the entorhinal cortex has been implicated as an important region of the brain that supports spatial navigation. Neurons within the entorhinal cortex exhibit activity patterns that are strongly correlated with the movement of the animal. These so-called Grid Cells maintain firing fields that are best described as triangular, repeating ‘tessellations’ across local space [14,15]. The entorhinal cortex hosts a variety of these activity patterns, with Grid Cell populations that have firing fields recurring at varying frequencies and each cell within that population representing a different subset of locations. Collectively, they are thought to form a metric representation for resolving problems with a structured spatial component; typically navigation in physical space, though there is evidence that Grid Cell-like activity is useful for ‘navigation’ in similarly structured, physical-analogous spaces, such as trees and relational graphs [16], as well as being found in brain regions far from the entorhinal-hippocampal complex [17].

A popular computational model of Grid Cell activity involves the so-called Twisted Torus model proposed by Guanella et al. [18]. Here, a population of neurons describe a conceptual 2D surface that forms an offset torus topology, closed in all directions; a kind of neural ‘Pac-Man space’. This space is realised via excitatory connections whose synaptic strengths encode a virtual distance between cells, with virtual neighbours having strong connections and virtually distant cells having weaker connections. Inhibitory influence can either be global, or follow the opposite rule to that of the excitatory connections; neighbours inhibit each other proportional to their ‘distance’.

This contrasts with the Oscillatory Interference model, whereby interference patterns between three intersecting theta waves can produce periodic firing fields [19,20]. Despite its elegance, it had been disregarded as a mechanism in part due to the lack of recorded theta rhythms in bats [21]. However, despite the absence of *continual* theta rhythms in bats, they do show activity in the theta band during navigation-relevant actions, such as active exploration of the environment through echolocation [22]. Furthermore, the presence of 3D place fields in bats [23] casts further doubt on the ‘all-in-one’ solution of the Twisted Torus, as the topology (connectome) would need to become even more complex. To represent 3D space using multiple ring attractor networks, as underlies the Oscillation interference model, would simply require the integration of additional ring attractors [24].

An early example of a biomimetic approach to robotic navigation is found in Milford’s RatSLAM algorithm [25]. Taking inspiration from the rodent entorhinal cortex, it models the robot’s 2D location (x,y) and head direction (θ) as an attractor state on a non-Euclidean, wrap-around 3D manifold. Odometry information is converted into velocity inputs that perturb the attractor state in proportion to how the robot moves. With all integral dead-reckoning mechanisms, the noise inherent in the velocity measurements accumulates, causing drift in the estimate over time. External sensory information that is anchored to an assumed static world can be used to correct for this drift by making the further assumption that drift earlier in the run is lower than later. This means that external sensory scenes observed in early ‘low-drift’ regions of an exploratory venture can be used to correct for drift in the dead-reckoning system when the robot later encounters the same scene. This implies the use of a memory system, which in RatSLAM is called the View Cell memory.

We place our model firmly in this arena, relying on the known existence of physical ring structures in invertebrate brains [1], neural homologies in the form of virtual rings in mammalian brains [2,3], and existing models showing the viability of periodic and ring-like structures in creating grid codes [19,20,24]. We forgo the explicit modelling of theta rhythms, including phase precession, in favour of focusing purely on navigation tasks and the use of realistic allothetic sensory cues in correcting for drift.

To collect this sensory data, we used the WhiskEye robot which was built to mimic whiskered mammals and equipped with both visual (cameras) and tactile (whiskers) sensors [26]. As seen in Figure 1, WhiskEye has been replicated in simulation and put within rich virtual environments, enabling the swift gathering of large visuo-tactile datasets using a model of tactile attention [27] to explore autonomously. These datasets have been used as training data for a Predictive Coding Network (PCN), a generative machine learning model that creates multisensory, entangled representations from two or more sensory modality streams. Further multisensory test sets can then be presented to the network and the generated representations gathered alongside odometry data, which is subsampled to find the position and orientation of the robot where the corresponding multisensory samples were captured. This gives a ground truth trajectory of (x,y,θ) poses and the corresponding sensory ‘experience’ at each pose [28].

In this study, we design a spiking neural network (SNN) model to track this trajectory. This takes the form of three SNN ring attractors, each integrating motion with respect to a principal axis and linked together by a population of Grid Cells. The ring state is mapped to conventional 2D Cartesian coordinates to give $\left(\hat{x},\hat{y}\right)$ estimates that can be compared to the ground truth trajectory. We demonstrate that this system can maintain an estimate of position that tracks the robot trajectory, can use multisensory experiences to correct for drift in this estimate and can do so at varying levels of power consumption. We also show that apportioning corrective input relative to the system’s confidence in its corrections has a positive effect on reducing drift. Lastly, we compare the performance of using visual, tactile and multisensory experiences as corrective inputs and demonstrate that the system is resilient to extraneous sensory modalities.

## 2. Materials and Methods

### 2.1. Ring Attractor Model

The Ring Attractors are modelled in NEST 2.18 [29], using an architecture based on prior models of the head direction system [30]. This involves an Excitatory Cell population to model the ring state, tracking one component of the trajectory; an Inhibitory Cell population to regulate the attractor dynamics, ensuring a unitary locus of activity within the ring; and two populations (+/−) of Conjunctive Cells, encoding incoming velocities by spiking proportional to the velocity vector’s alignment (+) or anti-alignment (−) to a particular ‘principal’ axis. Figure 2A shows how these cells are connected to form the ring structure. Preliminary work has suggested that three rings with principal axes offset $60^{\circ }$ from each other can produce Grid Cell-like firing patterns (Figure 2B), thus being able to track position in a 2D plane. Further details of this model are discussed below.

#### 2.1.1. Direct Inhibition of Conjunctive Input

Self-motion velocity inputs to the ring attractors are governed by Conjunctive Cells [15]. In the original head direction ring attractor model they were co-excited by both the excitatory manifold ‘below’, and by incoming angular velocity signals. Each excitatory neuron had two associated Conjunctive Cells, each of which is sensitive to either positive or negative angular velocity. Careful weight selection enabled them to act as a neural AND gate, contributing velocity signals only around the location of the attractor state, or activity ’bump’ as is often referred. Thus, inputs can only move the bump to adjacent cells, forcing the bump to sequentially traverse each adjacent neuron to reach a new resting state. This constraint grounds the ring attractor’s behaviour in the physical reality of head movement.

The problem we found in using this approach was that the maximum firing rate of the Conjunctive Cells was determined by the firing rate of the underlying Excitatory Cells, as they made the largest contribution to their activity. A better solution would be for the opposite to be true, whereby the firing rate of the Conjunctive Cells, and therefore change in the ring attractor state, would be predominantly proportional to incoming velocities.

To address this, the Inhibitory Cells were connected to the Conjunctive Layer with the same distance-dependent relationship as previous, replacing the direct Inhibitory-Excitatory connections. This directly inhibited most input to the network except for a small ‘window’ of non-inhibition above the ‘bump’. Inhibiting the input rather than Excitatory activity meant that the Conjunctive Cells could fire faster than their downstream Excitatory Cells, ensuring that rapid increases in velocity could be more accurately represented by a proportional increase in spike rate across a wider range.

However, this approach meant that the ring was entirely dependent on incoming velocity inputs to maintain its attractor state; a velocity of 0 m
s^−1^ would cause the bump to collapse. Clearly, it would be detrimental to lose path integration state every time the animal stops, which suggests that the ring attractor requires a ‘floor’ of constant velocity input to maintain its activity pattern. Recordings in vivo support this; cells that display Conjunctive Cell-like activity show a rate of spiking that, though proportional to velocity, does not diminish entirely when the animal is at rest [32]. This forms one half of the inspiration for the development of antagonistic input signals discussed in Section 2.1.2.

#### 2.1.2. Antagonistic Input Signals

One of the main causes of drift in the ring attractor came from rapid changes in velocity, particularly those that involve a change in direction. This change was resisted by the effective inertia of the bump making it difficult to represent some patterns of movement; the dynamics of the ring attractor not matching the dynamics of the agent’s movement. To help overcome this, an antagonistic mechanism was added to the input system:At rest, a baseline level of excitability is provided, with each Conjunctive population trying to push the bump left or right with equal intensity. This holds the bump ‘in tension’; ready to move, yet in balance.Changes in incoming velocity proportionally increase input to the population associated with its sign, with a equal and proportional reduction to the opposing population. Baseline input is increased proportional to velocity magnitude; speed coding [15].When a change in velocity direction occurs, the respective Conjunctive Cells are already close to the required spike rate to accurately convey the velocity information to the bump and induce appropriate movement.

This leads to the following calculation of the Conjunctive Cell inputs:$$\begin{array}{cc} C_{\min} & \in R^{+}\\ V_{\mathrm{mag}} & =||\Delta X(t),\Delta Y(t)||\\ V_{\mathrm{ang}} & =cos\left(atan2\left(\Delta Y(t),\Delta X(t)\right)\right)\\ V_{\mathrm{mag}}^{+} & =\{|v|\in V_{\mathrm{mag}}:V_{\mathrm{mag}}>0\;\mathrm{else}\;0\}\\ V_{\mathrm{mag}}^{-} & =\{|v|\in V_{\mathrm{mag}}:V_{\mathrm{mag}}<0\;\mathrm{else}\;0\}\\ C_{o}^{+} & =V_{\mathrm{mag}}-V_{\mathrm{mag}}^{+}\\ C_{o}^{-} & =V_{\mathrm{mag}}-V_{\mathrm{mag}}^{-}\\ C^{+} & =V_{\mathrm{mag}}+C_{\min}+V_{\mathrm{mag}}^{+}+C_{o}^{-}\\ C^{-} & =V_{\mathrm{mag}}+C_{\min}+V_{\mathrm{mag}}^{-}+C_{o}^{+} \end{array}$$
where ΔX and ΔY are N-by-1 matrices (arrays) of the agent’s velocity in the environment, $C_{\min}$ is the minimum velocity input, $C^{+}$ and $C^{-}$ are the positive and negative conjunctive velocity inputs and $C_{o}^{+}$ and $C_{o}^{-}$ are the opposites thereof. In this case, the opposite of *C* is its additive inverse, element-wise, such that a velocity of +2 would become −2 etc. This makes $V_{\mathrm{mag}}^{+}$ the set of all elements that are positive, with 0 replacing negative elements; $V_{\mathrm{mag}}^{-}$ is the opposite case, retaining negative elements only. This ensures that the $V_{\mathrm{mag}}^{+}$ and $V_{\mathrm{mag}}^{-}$ vectors retain the same length as their parent $V_{\mathrm{mag}}$. Taking the absolute for each *v* ensures they can be used equivalently in later steps. A velocity of 0 m
$s^{-1}$ at any point leaves only the baseline $C_{\min}$ to maintain the bump’s memory and allow for smooth integration of future nonzero velocities.

#### 2.1.3. Multiple Rings for Multiple Components

Grid Cells were first recorded in vivo within the entorhinal cortex of foraging rats [14]. Grid Cell firing fields tessellate the environment in a triangular pattern and are thought to contribute to navigational tasks, particularly to path integration [33]. Path integration appears to be computed by Stripe Cells [34], tracking movement along 3 axes offset at 60 degrees from each other.

Inspired by prior models, such as Kovacs [35], Horiuchi and Moss [24] and Burgess [36], we have built a 3-ring attractor system to model path integration using spiking neurons as media. Each unique combination of neurons between the three rings is connected to a Grid Cell, that will fire maximally when the activity bump of all three rings is centred on its connected ring neurons (see: Figure 2B for an illustration). This accommodates both the repeating nature of both Stripe and Grid Cells, implying a periodicity in their underlying substrate, and the known capability of biological neurons to form ring attractors.

In our model, each ring attractor is sensitive to a component of velocity with their principal axes at 0, +60, and +120 degrees offset from a globally anchored reference axis. Work by Mhatre et al. [34] has already shown how Spike Timing-Dependent Plasticity can lead to the emergence of Stripe Cells adopting this 60 degree offset from each other; we take this for granted and build our model with this relationship already established.

Parameters were hand-tuned to reflect the dynamics of the robot’s motion until the ‘bump’ was both stable at rest and moved smoothly to track velocity changes. Tuning was conducted through a piecewise simulation process, running the ring attractors for small timesteps, gathering data and viewing visualised activity of the three rings. This was combined with a view of the ground truth trajectory, plus the estimated trajectory from the ring activity, after unwrapping and transformation to the Cartesian coordinate system. An example of this visualisation is given in Figure 3. Note that the ground truth data was taken from simulation and thus has no noise by default; this is added post-hoc, as discussed in Section 2.6.

### 2.2. Predictive Coding Network

Derived from Dora et al. [37] and used to build representations for place recognition [28] and correct for drift in a head direction network [30], the MultiPredNet is a Predictive Coding Network (PCN) written in Tensorflow [38] that learns generative multisensory representations to predict its inputs across multiple modalities (vision and tactile). The PCN can also be run with one or more sensory streams disabled; the network will only encode information from active sensory streams, thereby enabling the generation of single-modality visual or tactile representations.

The PCN demonstrates that by training to predict incoming sensory inputs alone, representations suitable for place recognition can be learned. Indeed, separating incoming sensory inputs well necessitates the understanding that they occur under different conditions. As the environment explored is static, this leads to the robot’s own location and orientation being the primary factors that determine the nature of its sensory experience.

### 2.3. Associative Memory

To provide the ability to remember prior experiences, detect loop closures, and generate corrective signals to the ring attractors, we have built a dynamically allocated associative memory system inspired by RatSLAM [25] and developed from prior work [28]. This memory system takes as input the upper, multisensory latent layer of the PCN as the signature of an experience. Each incoming experience is then compared to all other previous experiences using Pearson correlation, where if the correlation between it and at least one existing experience is above a threshold (0.8) then the closest ‘memory’ is recalled. If however, the new experience’s correlation is below the threshold for all existing experiences—guaranteed if it is the first experience—then it is added to the collection of stored memories.

Each experience stored in memory not only holds the signature created by the PCN but also the current state of each ring attractor, that is, the location on the ring with the highest spike rate at that time. Upon a successful recall event, current will be injected into the 3 ring attractors at the stored targets, thereby adjusting the activity in each attractor towards the remembered state associated with that memory.

#### 2.3.1. Deriving Position from Ring Attractor State

Unwrapping the ring states causes them to act as pseudo-axes of a 3-value, planar coordinate system. The location in ’ring space’ can be transformed into Cartesian space by projecting perpendicular from each ring and finding the point at which these projections intersect. Once converted to Cartesian coordinates, the predicted agent location can be compared to the ground truth odometry data, enabling the performance of the ring system to be evaluated. The current model represents a spatial area of roughly 1.3 m${}^{2}$; all trajectories in this study stay within these bounds, meaning each ring state represents a unique location and does not require disambiguation with multiple spatial scales. This allows for the comparison of ring attractor state to ground truth trajectory as shown in Figure 3.

### 2.4. Compensating for Drift in the System

The ring attractor system models a given trajectory faithfully in the initial leg of the journey. However, relying on the integration of velocity changes alone leads to drift over time, a problem shared by all inertial navigation systems. Fortunately, the idiothetic navigation system represented by the rings need only be relied upon exclusively in conditions with poor access to familiar distal allothetic cues (be they visual, tactile, olfactory etc.) or when clear proximal cues (such as for determining translocation via optic flow) aren’t available. These condition are rarely long term and some form of external cue or memory trace will usually be available, even if only intermittently, to provide a known reference point that can be used to compensate for drift. This can be done by associating each stored memory with the current state of the idiothetic system (in this case, the ring state) and ’nudging’ the rings back toward that state upon recall.

To represent external corrective input from other brain areas, step current generators were connected to the rings. 3 rings of 120 neurons each gives a total of $120^{3}$ = 1,728,000 combinations of cells. To apply corrections faithfully to the rings, 1,728,000 separate injectors would be needed, which is infeasible both in terms of simulation size and neurophysiology. To reduce this complexity, we add in ‘Arc Cells’ that act as hubs for down-scaling the underlying ring activity, both for reading from the ring—to provide the ring estimate to downstream areas—and corrective injections, connecting across an exclusive patch of cells on the ring. To be clear, we are not predicting or assuming these cells exist in the entorhinal cortex; rather the influence of these Arc Cells would likely be realised as branches on the axonal or dendritic arbours of connecting cells. We choose to represent them as cells in NEST for ease of simulation. With 10 Arc Cells per ring the number of injection sites reduces to 1728, striking a balance between precision of correction and simulation resources.

In prior work [28], robot trajectories were gathered and sensory data, both visual (camera images) and tactile (whisker deflection values) data were fed to a predictive coding network (PCN). This PCN learned to abstract the two sensory streams into a compressed, 100-dimensional multisensory representation, for the purposes of predicting future sensory data in either stream. This representation also acts as a useful ‘signature’ of a multisensory experience, similar to the use of image signatures in RatSLAM. The metric of choice to compare memories is Pearson’s Product Moment Correlation Coefficient (PPMCC), as there has already been established a statistically significant positive correlation between the PPMCC of two representations and their distance in physical space [28].

As a corrective mechanism, these multisensory experiences can be stored alongside the current most active Arc Cell from each ring; the idiothetic coordinates of the experience. These can be stored as a combined ‘memory’ that can later be compared to other sensory experiences. When the current experience closely matches a stored memory, this suggests that the robot has returned to a location close to where it was previously. The ring state of that memory can then be used to adjust the current ring states, pulling their activity closer to where it was during the recalled memory. Figure 3 illustrates these injections taking place throughout the run and their effect on the rings’ collective position estimate.

### 2.5. Variable Power Consumption

Spiking neural networks (SNNs) are uniquely suited to robotic applications, tending to communicate sparsely and with low power requirements. This has lead to the development of neuromorphic computing devices [13], providing a target platform for these networks. Mobile devices, particularly those that require robustness and longevity in the field, would benefit enormously from the inherent redundancy and power efficiency of SNNs.

Further to this, a system such as a ring attractor model requires a minimum amount of energy to maintain ring activity, but beyond that, its performance will vary depending on the overall spike activity of the network—therefore the energy consumption—permitted. To evaluate this in this study, numerous trials of the system were conducted, varying the baseline amount of power provided to the rings.

### 2.6. Introducing Variance

The default model is entirely deterministic and has no stochastic components. To introduce variability between trials, the initial membrane potential of each neuron was randomised, and the velocity input varied by a small random value per timestep; for details, see Table 1. Each network was run 100 times with all seeds incremented by 1 after each run, with the exception of the Power Consumption experiment which was run 10 times per power value (4500 total). The Master Seed is used by NEST’s internal Lagged Fibonacci Generator [39], whereas the Membrane and Trajectory seeds are used by Numpy’s default Random Number Generator (PCG-64) [40], modifying the starting membrane potential and velocity inputs respectively. This process ensures that the model is robust to variation in the trajectory and does not rely on having a particular initial state. The datasets generated by this process are provided at https://github.com/TomKnowles1994/Biomimetics-Ring-Attractors.

### 2.7. Confidence

Using Pearson correlation to test the similarity of experiences also gives a continuous value that can represent the confidence of the system in that particular recall. This can be used as a scaling factor when applying corrective input; the rationale being that, if the current experience perfectly matches a past experience, the ring attractor state should be corrected maximally, as any deviation from the recalled state is assumed to be drift. For a less closely matching (but still above-threshold) recall, the correction will be proportionally weighted, as some of the deviation present may not be drift but representing an actual difference in current position from the recalled state.

### 2.8. Influence of Allothetic Sensory Modality

The relative contribution of visual and tactile features in the environment was also investigated. The network was run for 100 trials each for both visual and tactile corrections. These were then compared to the original (multisensory) corrected run and the uncorrected run, to see if lone modalities can improve network performance, how the performance compared to multisensory corrections and how visual corrections compare to tactile.

## 3. Results

### 3.1. Spiking Ring Attractors Can Track Position

Once unwrapped and converted into Cartesian coordinates, as discussed in Section 2.3.1, the ring values provide an estimate of the agent’s position. This permits comparison to the ground truth position and the measuring of performance. The rings adjust this estimate consistently with incoming changes in velocity, even when corrective influences are not provided. However this estimate is subject to drift over time, with the estimated position deviating considerably by the end of the trajectory.

### 3.2. Spiking Ring Attractors Benefit from Corrective Input

Using earlier memories of both the environment and the ring state can correct drift in the ring state and thus improve the accuracy of location estimate. Extensive testing found that the optimal corrective input strength was 450 pA and the optimal corrective input duration was 500 ms. Smaller, shorter current injections failed to influence the ring state proportional to velocity inputs, whereas larger, longer injections would disrupt bump stability. Applying these corrections lead to a significant drop in error across the run, as seen in Figure 4.

### 3.3. Higher Power Consumption Reduces Error

The stability of the ring activity and the consistency of its position estimate is proportional to the amount of energy used to maintain it, with a robust, graceful decline in performance as current is reduced. Similarly, a larger energy input results in more spikes per unit time which can both excite and inhibit the network, such that it more faithfully describes the trajectory of the robot. This comes with 3 caveats:A certain amount of current is required to establish the attractor dynamics and generate any bump at all; currents below this fall into the ‘dead zone’The error declines proportional to input current for a certain range of values onlyPast a certain value, extra input current has an inconsistent effect on performance

The dead zone arises due to the dependence of the ring attractors on their conjunctive input; below a certain threshold (approximately 500 pA), too few spikes are generated to overcome the latent inhibition in the network. The increase in performance is due to increased responsiveness of the conjunctive cells; with a higher baseline input provided to both populations, the bump activity is better able to represent large accelerations and thus more faithfully model the robot’s motion. However, a too high baseline input disrupts the ring attractor dynamics through over-excitation. The ring system was run for 10 trials for each input current value, incremented by 10 pA each time. This demonstrates the reliability of the effect across different runs and current values, preserving the accuracy-power relationship despite the variance discussed in Section 2.6. Full results are shown in Figure 5.

### 3.4. Correction Intensity as a Proxy for Confidence

As the strength of recall is variable between recall events, it leads naturally to the question as to whether making corrections proportional in intensity to this recall strength would effect the ability of corrective inputs to compensate for drift. Figure 4 shows how scaling the amount of current by the recall strength leads to a better corrective input and therefore further reduced drift.

### 3.5. Sensory Modality of Memories Affects Corrective Performance

Using visual, tactile and multisensory experiences from the PCN as corrective inputs affect the mean error in different ways. Figure 6 shows significant improvement when using both multisensory and visual experiences, but not tactile. The benefits of multisensory vs visual experience are not significantly different, suggesting that the contribution from the tactile component of the multisensory experiences is minor—though it is worth noting that the addition of tactile data does not confound the multisensory experiences. This suggests that the experiences encoded by the PCN are resilient to extraneous sensory modalities, and thus useful in highly heterogenous environments where the utility of different sensory modalities changes rapidly.

## 4. Discussion

This study has demonstrated that coupled ring attractors implemented as SNNs are able to serve as a reliable and robust navigational aid for a robot. Their combined activity mimics that of Grid Cells, not only in the visual similarities of their firing patterns, but also in their cyclical representation of position and their propensity to be influenced by multisensory memories of the external environment. In support of this, several improvements have been made to prior SNN ring attractor models with success in modelling head direction being naturally extended to success in modelling planar location, leaving scope to develop this into a fully spike based estimate of (x,y,θ).

The ability to vary the baseline input to the ring attractors makes for a flexible path integrator, one that can trade drift accumulation for power saving and vice versa. This allows for neuromorphic systems with intelligent power usage, with the network sustained by a lower, baseline level of power when corrective sensory inputs are readily available, but increasing power consumption, and reducing the inherent drift, when sensors fail or become uninformative. This capability could find use in applications where sensory data is intermittently available, such as navigating mines and tunnel systems, and where graceful degradation in performance is required to manage a degrading or limited power supply, such as for space exploration and field robotics.

The utility of weighting corrections by recall strength exploits the similarity of memories in ‘experience space’ to correct for discrepancies in physical space, demonstrating the close mapping of non-spatial experiences of a rich environment to explicitly spatial data. It is worth speculating if this relationship can be reversed; if indeed the inherent physically-congruent activity of ring attractor system can be used to scaffold the organisation of non-spatial information in the brain, with ring attractors structuring sequences or even ‘spaces’ (or palaces [41]) of memory to be recalled. These ideas, of separating memories in cortex as they are separated in space, echo ideas of Buzsáki and Moser [42].

The superior performance of visual data in providing corrective input agrees with prior work [28], where visual and multisensory representations correlated better with location than tactile. This may be due to the utility of visual cues for localisation—being able to acquire useful stimuli at both close and long range—or to the lack of contact time the whiskers had with objects. Future datasets from a more densely packed environment, or with a greater variety of object shapes and sizes, may yield better results for tactile corrective input.

Several areas of this model mimic features of entorhinal cortex processing but do not implement it in spiking neurons. These include:The corrective influences being injected as step currents, rather than spike trainsThe Conjunctive Cell inputs being pre-calculated rather than being mediated by other documented neuron behaviours, such as using Head Direction Cells [43] plus orientation information from the retrosplenial cortex [44] to apportion velocity componentsA bio-plausible model of memory that can be used to trigger corrective inputs.

For (1), a Poisson generator could substitute in the short term, or a new model of memory recall created, perhaps using bio-plausible pattern-completion circuits such as proposed in the dentate gyrus [45]. This is related to (2), requiring additional spiking models to describe the functionality of upstream brain areas. For (3), a system similar to that of Ocko et al. [46] using Landmark Cells to mediate recall could be investigated; in this case, they would be ‘Experience Cells’ working with landmark-free representations.

This model is also straightforward to extend to three or more dimensions, with bat-inspired models of 3D navigation systems based on ring integrators showing Grid Cell-like fields in three dimensions [24] with the addition of a fourth ring. This would make the model suitable for environments where the vertical dimension is critical, such as aerial, submarine or subterranean environments. Going beyond this to 4D or 5D navigation problems is nonsensical in terms of purely physical space, but could be useful for adding non-spatial context cues to navigation problems, such as modelling seasonal cycles or the rhythm of a particular task [47].

Regardless of specific improvements, it is clear that an effective brain-inspired navigation system involves an entire complex of interacting circuits to be effective. This is to be expected, given that, in the Mammalian brain, the entorhinal cortex is strongly interconnected with other areas. So far, this system has been demonstrated using set of trajectories derived from one robot using hand-tuned parameters; an interesting further study would be to investigate a bio-plausible Hebbian learning approach, with bidirectional calibration between other areas in the hippocampal complex for a self-adaptive learning system. In addition, the role of multi-scale Grid Cell populations, as first observed by Stensolla et al. [48] within rat entorhinal cortex, would also be a natural extension to this work through simple duplication of the proposed coupled ring attractor architecture, enabling trajectories spanning larger areas to be represented with the same precision.

## Figures and Tables

**Figure 1 biomimetics-08-00399-f001:**
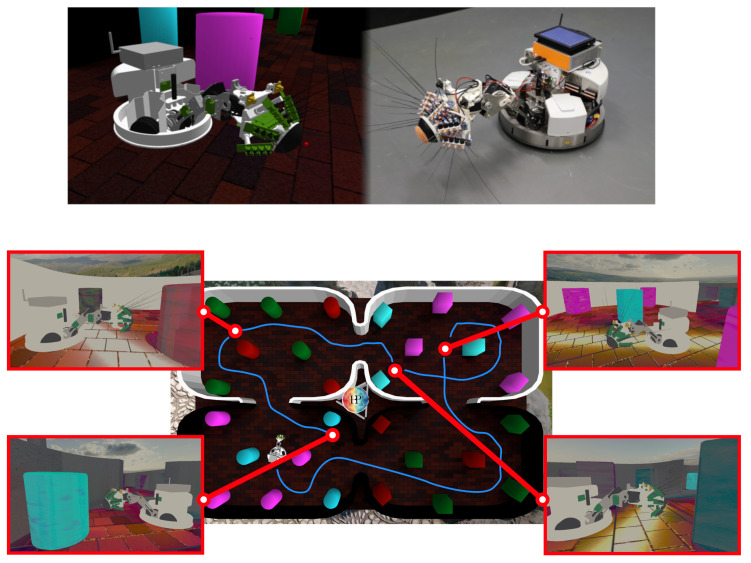
WhiskEye is a biomimetic robot platform that mimics the sensory systems and behaviour of rats and shrews. It has been recreated in simulation along with rich visuo-tactile environments. Exploring these environments autonomously, it gathers multisensory datasets along with odometry data describing its trajectory. These ‘experiences’ of the environment are later used as corrective inputs.

**Figure 2 biomimetics-08-00399-f002:**
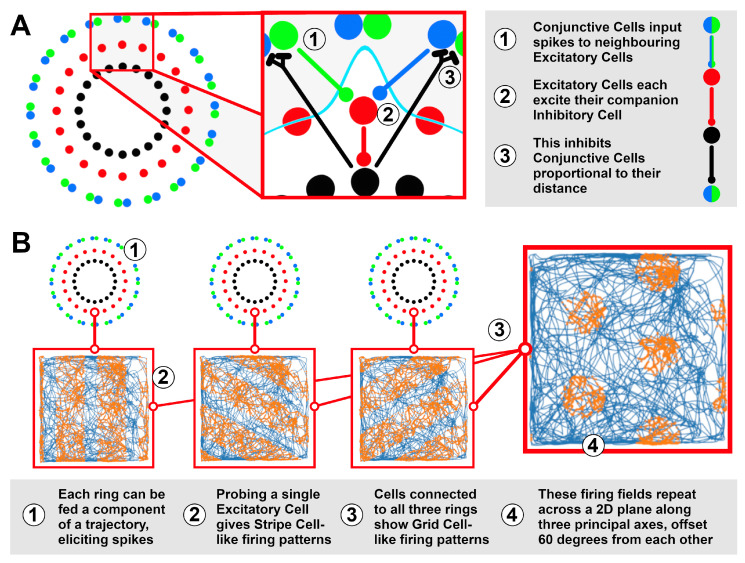
(**A**) The ring model is composed of repeating units of Excitatory, Inhibitory and Conjunctive Cells that work together to integrate an input into a unitary ring state. (**B**) A single ring integrating a component of a trajectory (blue trace, from [15]) will have its Excitatory Cells spike (orange) in regular intervals along an axis, similar to recorded Stripe Cells [31]. A cell taking input from three rings, each integrating a different component of the same trajectory, produces Grid Cell-like firing patterns, suggesting that three rings are collectively able to track position in a 2D plane.

**Figure 3 biomimetics-08-00399-f003:**
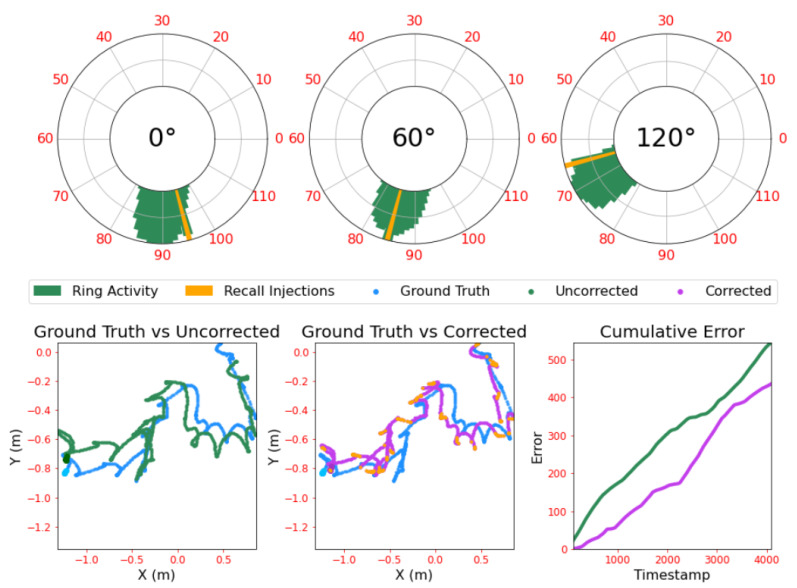
An example of the visualisation used for the ring tuning and evaluation process, with ring principle axes being aligned to $0^{\circ }$, $60^{\circ }$ and $120^{\circ }$ offsets from the reference axis as described in Section 2.1.3. Polar plots represent the three ring attractors with ring activity composed of spikes gathered over a 100 ms window, with their unitary activity bumps clearly visible. Trajectory plots show the ground truth trajectory the rings are trying to track, with the Cartesian transformation of the ring states overlaid as per Section 2.3.1. The error plot shows the cumulative error of the uncorrected and corrected ring attractor estimates, with each timestamp representing 20 ms of simulation time. An animated version of this figure is provided at https://github.com/TomKnowles1994/Biomimetics-Ring-Attractors.

**Figure 4 biomimetics-08-00399-f004:**
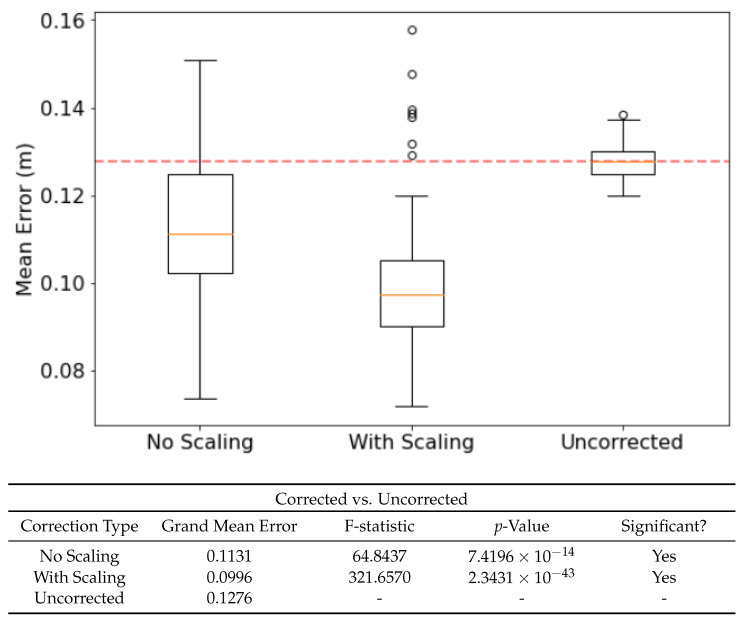
The effect of ‘confidence scaling’ the injected currents on mean error. When enabled, the injected current will be proportional to the Pearson’s Correlation between the incoming sensory view and best-matching recalled memory. F-statistic and *p*-value taken from an F-test (n=100, α=0.05). $C_{\min}$ was set to 2700.

**Figure 5 biomimetics-08-00399-f005:**
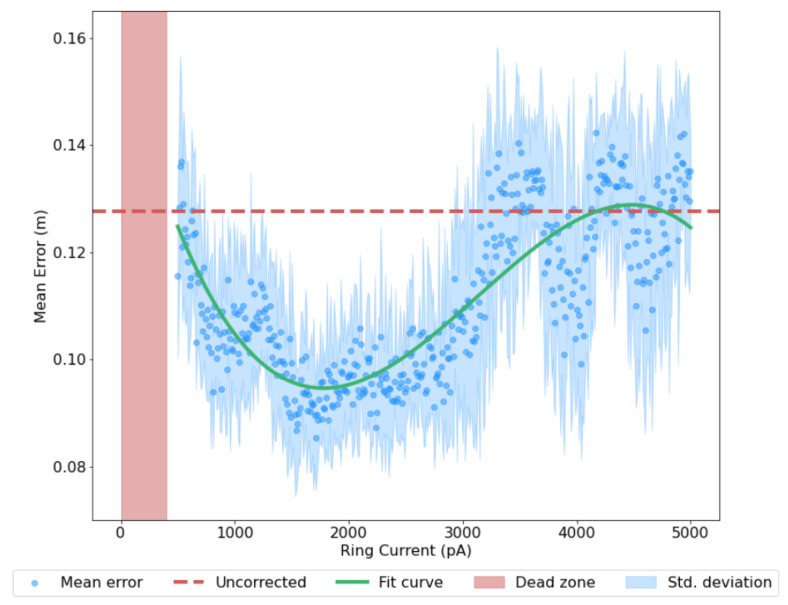
The effect of varying baseline current input $C_{\min}$, a proxy for power consumption, on mean error. Test values increment by 10 pA with 10 runs conducted for each value. Error is given as the cumulative Euclidean distance between the ground truth trajectory and the corresponding ring state-estimated trajectory. The dead zone represents no activity in the ring, due to a lack of sufficient input current. Confidence Scaling applied as per Section 3.4.

**Figure 6 biomimetics-08-00399-f006:**
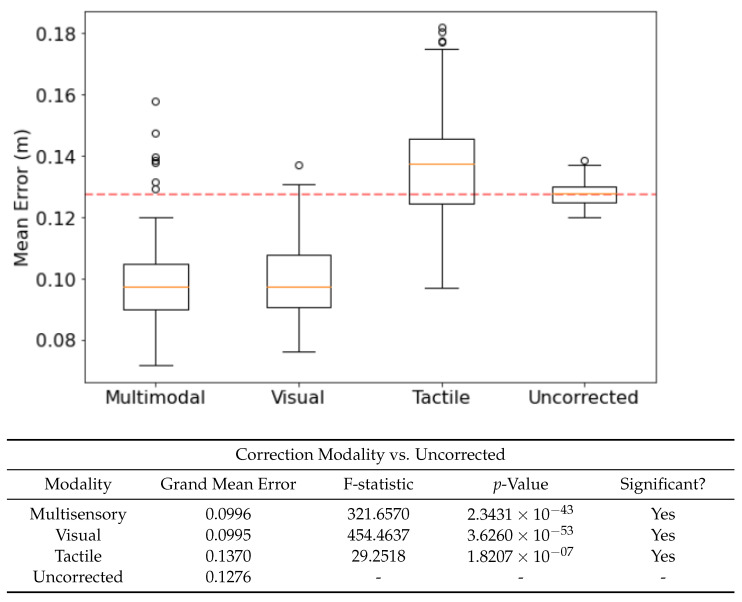
The effect of correction modality—the sensory data used to form the experience memories—on performance, benchmarked against the uncorrected run. Confidence Scaling applied as per Section 3.4. F-statistic and *p*-value taken from an F-test (n=100, α=0.05). $C_{\min}$ was set to 2700.

**Table 1 biomimetics-08-00399-t001:** The parameters for the Random Number Generators used in the simulations. After each condition was tested for its requisite number of runs, the seed would be reset to its Seed Start value.

RNG	Seed Start	Distribution	Mean	Variance
Master	9032867582	-	-	-
Membrane	2390786556	Uniform	-	[−70, +55]
Trajectory	6983476394	Normal	0	0.1

## Data Availability

Code and data used in this paper have been made available at https://github.com/TomKnowles1994/Biomimetics-Ring-Attractors.

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
