# Peer review of "Ring Attractors as the Basis of a Biomimetic Navigation System"

_biomimetics, 2023, doi:10.3390/biomimetics8050399_

Round 1

Reviewer 1 Report

Please find it as attached.

None

Reviewer 2 Report

1.Why should we model ring attractors in 2.1 and what are the important subsets?

2.What are the advantages of using Cartesian coordinate system conversion in 3.1 and the differences before and after conversion?

3.What is the relationship between Pearson correlation coefficient and its physical space in 2.4?

4.Why is variance introduced in 2.6? The cited data is not provided.

5.Why are errors reduced in 3.3 and not expressed clearly?

6.What are the advantages of triangles in the description in 4.2?

Reviewer 3 Report

This paper proposed an idea to provide navigation ability to a robot by using Spike neural network. The approach can be used as a supplementary navigation in where power supply is limited, and external sensor is not providing reliable navigation information. This topic is interesting; however, this paper didn’t show a promising way to achieve it in real life. Some comments are listed below.

1.    The motion is predicted by a 2D model, does it work for a 3D occasion. What research is further needed for a 3D navigation?

2.    What kind of real application are this research for? How is the performance of current application status?

3.    In line 78-79, the author mentioned taht ‘the noise inherent in the velocity measurements accumulates, causing drift in the estimate over time’ If the velocity measurement is not stable or reliable, how to make sure your data is good to do navigation.

4.    The author tend to use only one sentence to describe the figure, and put a lot of details in the description of the figure. The writing style is not easy for readers to understand correctly.

5.    In figure2, there is no legend for the colors and shapes, it is not easy to understand as an overview.

6.    The ground truth of the PCN comes from the sensors, what is the sensitivity and response time of the sensors?

7.    The SNN is designed to be used in limited power environments, did you take the power assumption of the training, camera, sensors, etc? In my perspective of view, it doesn’t sound power effective.

The way of writing is not easy for reader to understand. For example, there is no description and anlysis for the figures in the main text, instead the author write part of the information underneath the figure. 

For the figure, there is no legend or discription about different shape and color, which is hard to know what information the author want to convey. 

Round 2

Reviewer 3 Report

This version looks good to me, can be pulished as it is. 

Author Response

We thank the reviewer for their comment and for all of their feedback thus far. It has been a great help to improve the quality of the manuscript.